# Enigmatic Formations Found in Routine Orthopantomography (OPG) Examinations: A Case Report

**DOI:** 10.3390/diagnostics13050840

**Published:** 2023-02-22

**Authors:** Riccardo Nocini, Luca Sacchetto, Morris Zarantonello, Alessia Pardo, Michele Bonioli, Daniele De Santis

**Affiliations:** 1Section of Otorhinolaryngology, Head and Neck Department, University of Verona, 37129 Verona, Italy; 2Section of Oral and Maxillofacial Surgery, Head and Neck Department, University of Verona, 37129 Verona, Italy

**Keywords:** salivary glands, scialography, medium contrast, CBCT

## Abstract

We describe two clinical cases of occasional radiographic findings on orthopantomography (OPG) that were performed routinely, for which the definitive diagnosis may be uncertain. After an accurate remote and recent anamnesis, for reasons of exclusion, we hypothesize a rare case of the retention of a contrast medium in the parenchyma of the major salivary glands (parotid, submandibular, and sublingual) and their excretory ducts as a consequence of sialography examination. In the first case we analyzed, we found it difficult to classify the radiographic signs on the sublingual glands, left parotid, and submandibular, while in the second case, only the right parotid was involved. Using CBCT, the spherical findings were highlighted, with multiple having different dimensions, as well as radiopaque in their peripheral portion and more radiolucent inside them. We could immediately exclude salivary calculi, which usually have a more elongated/ovoid shape and appear homogeneously radiopaque without radiolucency areas. These two cases (of hypothetic medium contrast retention with unusual and atypical clinical-radiographic presentation) have very rarely been comprehensively and correctly documented in the literature. No papers have a follow-up longer than 5 years. We conducted a review of the literature on the PubMed database, finding only six articles reporting similar cases. Most of them were old articles, demonstrating the low frequency of this phenomenon. The research was performed using the following keywords: “sialography”, “contrast medium”, “retention” (six papers) and “sialography”, and “retention” (13 papers). Some articles were present in both searches, and the really significant ones (defined after a careful reading of the entire article and not only of the abstract) resulted only in six occurrences in a time span from 1976 to 2022.

## 1. Introduction

The use of radiographic aids in clinical practice is essential (most of the time) to access a correct diagnosis, allowing staff to highlight structures and anomalies and especially hard tissues that otherwise are not identifiable with only instrumental and visual investigations [1,2,3,4]. The interpretation of the radiographic image is not always without errors and mistakes [5,6,7]; sometimes, a correct interpretation is difficult, making a correlation of clinical and anamnestic findings difficult also. Originally, in order to add information regarding glandular tissues in the orofacial area, an OPG examination was performed, and this was consequent to an injection of radiopaque liquid that allowed the parenchyma to be better visualized. Sialography (or scialography) is a radiological technique that is rarely used today, which consists of injecting an iodized and radiopaque contrast liquid/media by a blunt needle or a thin catheter into the outlet of the salivary duct (Stensen, Warthon, Bartolini) and is subsequently used in the execution of traditional radiographic examinations, such as OPG or computerized tomography, with the aim of visualizing the salivary canal, the gland’s parenchyma under examination, and any related problems in detail. Used since the mid-1920s [8,9], this technique has undergone numerous modifications and improvements [10,11,12,13], but currently, due to its local invasiveness, it has been almost completely replaced by echography.

In the current literature, there are few articles reporting analogous events. In our review of the literature on PubMed (as previously described in the abstract), only six significant articles were selected over a time span exceeding 35 years (1976–2022). In order to further support the importance of the cases that we will shortly describe, we found that such a small number of papers can be linked to the fact that, generally speaking, the more one goes back in time, the fewer published articles can be found, while nowadays, the number increases exponentially yet the rarity of the subject provides very little documentation that is correctly described. The six articles mentioned above are summarized in the table below (Table 1).

The first paper from El-Hadary et al. (1986) represents a case in which the prolonged retention of an oil-based contrast medium was recorded, most likely due to the inadvertent placement of the contrast medium into the extraglandular/extraductal tissues of the parotid gland [14]. N.S.M., a 40-year-old Saudi female, was seen on 12 January 1981 for routine dental treatment. As part of the diagnostic work-up, an OPG was taken and revealed multiple irregular radiopaque masses that were considerably anterior to the normal location for a parotid sialogram and were widely dispersed. The patient admitted that in August 1980, she had been referred to a hospital for a left parotid sialogram as part of an investigation into a recurrent swelling over a six-months duration. The sialogram was reported as a normal parotid gland, although the emptying phase was not mentioned. The patient received routine dental care but was advised that there was no indication of active treatment for the retained contrast medium. In January 1985, further radiographs revealed that although the contrast medium was still present, it had decreased considerably in both amount and the area covered. The symptoms were not related to the retained contrast medium, and no treatment was indicated, particularly since there was now a decrease in its extent.

In a report by Shigetaka et al. [15] (1996), a 69-year-old woman visited their clinic on 6 February 1992 with a complaint of buccal swelling and trismus. The symptoms had begun 1 month before. Two and a half years before, she had undergone sialography for her right parotid gland with Lipiodol Ultra-Fluoride (Laboratoire Guebert Co, Paris, France) in the otolaryngology department for a right buccal swelling. The doctor’s record contained no specified diagnosis. She was not treated at that time, but she had shown no further symptoms for approximately 2 years. The examination performed on 6 February 1992 showed severe swelling and erythema of the skin of the right cheek and the parotid region. The palpation showed marked tenderness, and the area felt warm. On a plain radiograph, there were many spherical radiopaque areas seen in the right midface region. A CT scan showed a soft mass lesion containing radiopaque areas of various sizes in the right cheek, the parotid gland, and the pterygomaxillary region. Based on the history and clinical and radiographic findings, a diagnosis of an inflammatory reaction due to exogenous Lipiodol was made. An intraoral biopsy was performed, and the condition was diagnosed as lipogranuloma.

More similar to our cases is the report by Ozdemir et al. [16] (2004); sialography was performed in a 23-year-old woman using Lipiodol Ultra-Fluide (Tamac, Turkey) to investigate the cause of a painless swelling in the region of the left masseter. After investigating her history and following a clinical examination and sialography, the patient was diagnosed as having muscle hypertrophy due to a one-sided chewing habit. Three years later, an OPG was taken and demonstrated small spherical radiopacities in the vicinity of a Stensen’s duct on the left. There were no complaints related to this lesion in the patient.

Another single case report was described by Macan et al. (2007) in which iodinated oil (lipiodol ultra-fluid (UF) leaked from an iatrogenic perforation of a Stensen’s duct and constituted a foreign body in the cheek; in the two years following sialography, the UF was still not being totally resorbed [17]. No radiological signs of reactive inflammatory changes to the soft tissue were observed, and the patient (like in our two cases) did not feel pain nor personally detectable swelling or achiness in the area, which represents a typical occasional routine finding [4]. At the time, they believed the contrast agent arrived beneath the skin but external to the platysma through a simple perforation in either the duct and/or the mucosa [17].

The only study with a larger number of patients is Schortinghuis et al. [18] (2009), which assessed the prevalence of lipiodol retention after parotid sialography and determined if the retention of lipiodol was related to the sialography technique or any underlying salivary gland pathology. When using the electronic hospital database (1996–2006), 66 out of 565 patients were identified who had additional maxillofacial radiographic examinations after the initial sialography. Additional radiographs up to October 2007 were included; these were OPG radiographs in all cases. In 28 patients (42%), signs of lipiodol retention were observed (mean radiographic follow-up: 15 ± 13 months). Retention was characterized by small radiopaque spots in the periphery of the gland. Lipiodol retention was predominantly associated with a fausse route (*n* = 8) or the presence of salivary gland disease (sialectasia; *n* = 17). In nine patients with signs of lipiodol retention, a series of radiographs were available. Lipiodol radiodensities decreased in size over 28 months and could disappear gradually (follow-up of 14–57 months). Despite the high frequency of retained small depots of lipiodol for years after sialography in patients subjected to additional radiographic examinations, no clinically adverse effects were observed.

Schilt et al. [19] (2016) cured a 49-year-old man of swelling in the right cheek after a parotid sialogram. The sialogram was performed using 2 mL of Ethiodol under fluoroscopic guidance. The fluoroscopic X-ray showed no ductal or parenchymal filling, but the immediate and unexpected pooling of contrast had occurred in the buccal soft tissues. At that point, the procedure had been terminated. The dehiscence or disruption of the Stensen’s duct with Ethiodol extravasation was the presumed diagnosis. In order to remedy the contrast extravasation, the patient was placed on oral cephalosporin antibiotics, and the body’s reaction to the contrast was clinically monitored. Within 10 days, several localized collections appeared within the soft tissues of the cheek. After 3 weeks, these increasingly well-demarcated sites became abscesses and were drained. The drainage demonstrated an oily content without a purulent component and appeared to be Ethiodol having migrated to the skin surface. The soft tissues healed completely without further abscess formation or symptoms over an 11-month follow-up period.

A truly peculiar thing about our case repot is that, at the moment, there has been no confirmation in the literature regarding the retention of a contrast medium that has remained in place for at least 20 years, a much longer time than all the cases described previously.

## 2. Case Presentation

In the first case CV, a 77-year-old Italian female patient was sent for a specialist visit at the Head and Neck Department of the University Hospital of Borgo Roma—AOUI of Verona, Italy, for investigations following enigmatic findings on OPG that were not attributable to radiographic artifacts. Her anamnesis includes a neck echography from December 1990, in which was found a “*welling in the left latero-cervical region, expansive formation with ovoid morphology located in the superficial muscle planes with longitudinal major axis of 15 mm and thickness of about 5 mm. This expansive formation is uniformly hypoechoic, spatially well defined, compatible with fibrolipoma*”; this was removed in June 1991, with a biopsy report of “*fibrous blood material incorporating inflammatory granulation tissue with hemosiderin deposits*”. A few years later, the patient underwent a new operation in the same site that found “*fibro adipose fragment and partly skeletal muscle with hemorrhagic infiltration area associated with reparative phenomena in fibrous evolution. nerve trunks with proliferative changes such as amputation neuroma and metaplasic bone nodule*”.

At the time of the visit in February 2021, the patient was in good general health conditions, denying drug allergies or pathologies, with no pharmacological therapy in place. On extra-oral physical examination, no swelling or asymmetries were evident. No problems were reported upon inspection of the soft and hard intraoral tissues. Upon palpatory examination of the sublingual and submandibular glands, harder round masses (than the surrounding tissues) were not evident with clarity and continuity. Examining the OPG (Figure 1) and Cone Beam Computed Tomography (CBCT) exam (Figure 2, Figure 3 and Figure 4), the spherical findings were highlighted, with multiple different dimensions, radiopaque in their peripheral portion, and a greater radiolucent inside them. which was in accordance with the anatomical localization of the major salivary glands (parotid excluded). These unusual and rare findings, after the careful exclusion of numerous pathologies involving the major salivary glands, were hypothetically interpreted as radiographic signs of residues of contrast medium used in the sialography examination, which the patient incidentally remembered undergoing many decades prior.

Different slices of CBCT (Cone Bean Computer Tomography) of CV patient from a different prospective.

In the second case (MR), a 67-year-old Italian female patient went to a private dental studio in Badia Polesine (Rovigo) for a routine check in mid-January 2021. Her remote and recent medical history revealed hypertension compensated for with medical drugs, previous mild episodes of gastritis, and the use of statins for cholesterol balancing. During the visit, the patient did not describe any problems of dental interest. Upon inspection of the perioral and extraoral soft tissues, small masses with smooth surfaces of varying sizes were found on palpations in the thickness of the right cheek in the subzygomatic area, which was not painful, if not under extreme compression. No decrease in parotid function was reported or noted. In order to deepen the investigation and complete the visit, OPG was performed (Figure 5) and revealed the appearance of multiple rounded masses mixed with radiopaque radiolucent. At this point, further questions were asked regarding any specific examinations of the salivary glands performed in the past, and we were told that a sialography was performed (according to the patient) at least 20 years ago, which she had partially forgotten. It was not possible to trace further information regarding the timing or method of the alleged and hypothesized sialography. At the specific request of the patient herself, for in-depth diagnostics, a low-dose CT scan was performed. It showed these masses with higher resolution in terms of their location and internal/external features (Figure 6, Figure 7 and Figure 8).

Unfortunately, it was not possible to obtain a longer follow-up of the case and document it in more depth from a clinical point of view due to the hospitalization of the patient, who deceased in October 2021 due to complications related to lung cancer.

Different slices of CBCT in second patient MR.

## 3. Discussion

Both patients referred to the absence of pain or other symptoms in the maxilla-facial area. By using two-fingers, the manual palpation could be detected on the regular surface masses in the right cheek in the subzygomatic area in MR; this was less noticeable in the case of CV.

In the first case, OPG and computerized tomography showed round masses of between 2 and 11 mm in diameter, anatomically resulting in the left parotid salivary gland and in the left submandibular gland; the masses on the sublingual gland area were less easily identifiable due to the overlapping of the chin portion and its thick cortical, but these were better detectable through the multiple slices of CBCT (Cone Bean Computer Tomography) than with different perspectives.

In the second case, the OPG radiographic examinations showed multiple rounded masses, from 3 to 6 mm in diameter, which were more radiopaque in their peripheral portions and more radiolucent in the central areas, anatomically compatible with the accessory parotid gland. The path of the Stensen’s duct is not in evidence. Computerized Tomography allowed us to visualize more details about these enigmatic formations (Figure 6, Figure 7 and Figure 8). In this case, there were about a dozen lesions clustered in a smaller space than in the previous case, with a rounded/ovoid appearance, markedly more radio-opaque in their irregular external perimeter, and radiolucent in their nonhomogeneous inner part. They appear to have dimensions ranging from a few millimeters to about 6.2 × 5.2 for the largest lesion, which is visible more clearly in the OPG and in the CBCT horizontal slice (Figure 7).

The major salivary glands and the submandibular, parotid, and sublingual glands play an important role in preserving the oral cavity and dental health [20,21,22]. Patients with problems of the major salivary glands may present various symptoms, such as dry mouth, an obstruction of the duct with salivary calculi and/or ductal strictures, dysphagia, and inflammation. Imaging plays an important role in visualizing the morphology and function to establish a diagnosis for treatment and for surgical planning. There are several options for diagnostic imaging: plain radiography, sialography, ultrasound (US), magnetic resonance imaging (MRI), computed tomography (CT), salivary gland scintigraphy, and F-FDG positron emission tomography (PET) [23]. Sialography is a radiological investigation that was used much more frequently in the past to examine various pathologies of the major salivary glands [24,25,26,27,28].

Once the contrast agent has been inserted into the duct and salivary glands to be examined, the reflux of the same area generally occurs in around 5 min. Prolonged retention, on the other hand, is considered to be an indication of reduced glandular function. Qwarnstrom noted that even with a normal gland, there is some transient alteration in gland function with sialography [29], with some authors not having clinical evidence of the permanent manifestation following such prolonged retention or not including the complications of the procedure [30,31,32,33].

A delayed excretion contrast medium into the surrounding connective tissue with pronounced parenchymatous filling can occur with both viscous oil-based media [34,35] or water-based media [36,37].

In our case, the residues of the contrast medium were found years/decades after the date of the diagnostic test, which makes it peculiar among the very modest number of case studies in the literature. Upon physical examination of the two patients, they showed no particular signs or symptoms, reporting (in their opinion) that they were in excellent health and that they had gone to the dentist for a routine check-up. Once again, patients do not feel any burning sensation in the mucous membranes, or even any difficulty in chewing/swallowing solid food; hence, we assume the absence of functional problems in glands, or a salivary compensation from the contralateral salivary gland. During the two-fingers palpation exam, a smooth and solid sensation of multiple rounded masses was found, confirmed by the radiographic exam from which masses from 3 to 7 mm in diameter can be identified, with more radiopaque in their peripheral portions and more radiolucent in the central areas. This specific radiographic connotation allows us to exclude that these signs are indicative of salivary calculi, which usually have a more elongated/ovoid shape and appear homogeneously radiopaque without radiolucency areas [38,39,40,41,42,43].

Few cases remain correctly documented in the literature of permanent residues of contrast medium into the major salivary glands [16,18].

## 4. Conclusions

Our review of the literature so far (mid-July 2022) suggests that these occasional findings are extremely rare in clinical practice, and because of a lack of specific symptoms and basically no characteristic appearances, its clinical diagnosis may be challenging. No follow-up over 5 years has been reported, unlike our cases (more than 20 years). This article is intended to help clinicians during the diagnostic phase who face similar radiographic findings, which are atypical, unusual, and little known, producing, in some cases, an incorrect diagnosis and treatment plan or over-treatment. We underline again how these types of radiographic investigations are in complete disuse by now, which is why such occasional findings are to be considered clinically rare, and they should be known by the clinician.

## Figures and Tables

**Figure 1 diagnostics-13-00840-f001:**
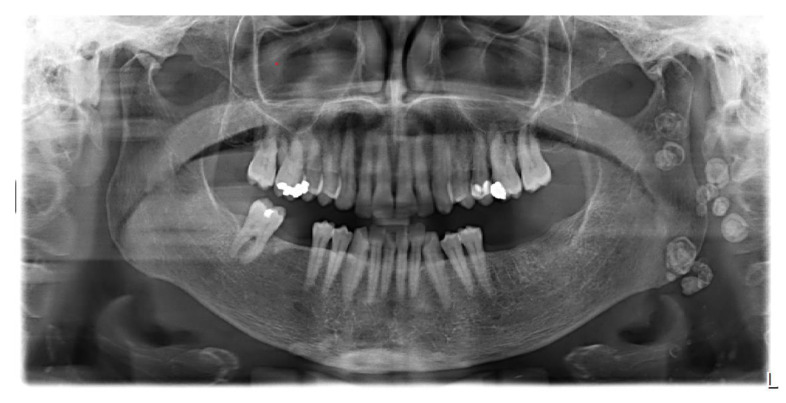
Orthopantomography of first patient CV, in which irregular round masses could be observed in the left parotid salivary gland and in the left submandibular salivary gland.

**Figure 2 diagnostics-13-00840-f002:**
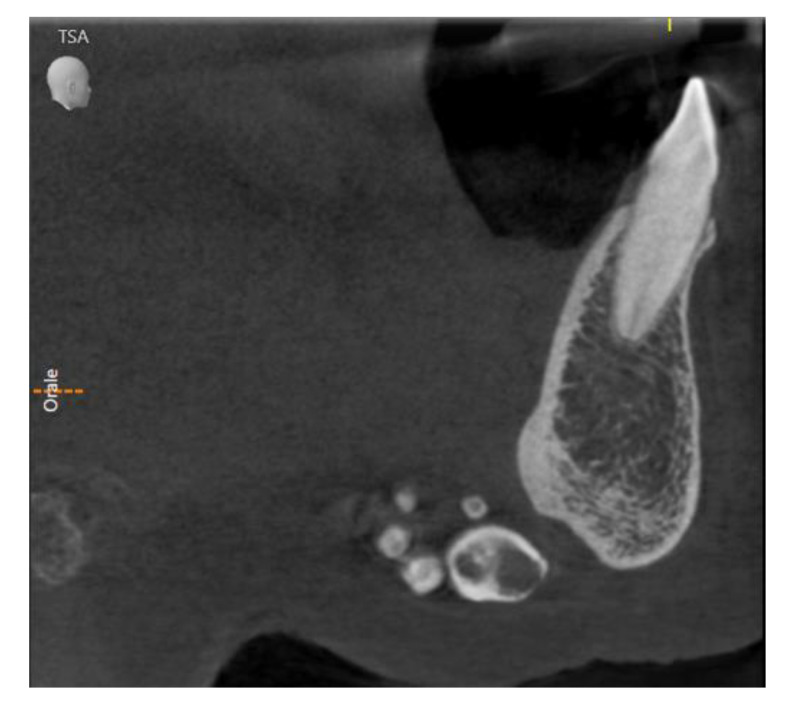
Sagittal slice of CBCT in sublingual salivary gland area, showing multiple round masses.

**Figure 3 diagnostics-13-00840-f003:**
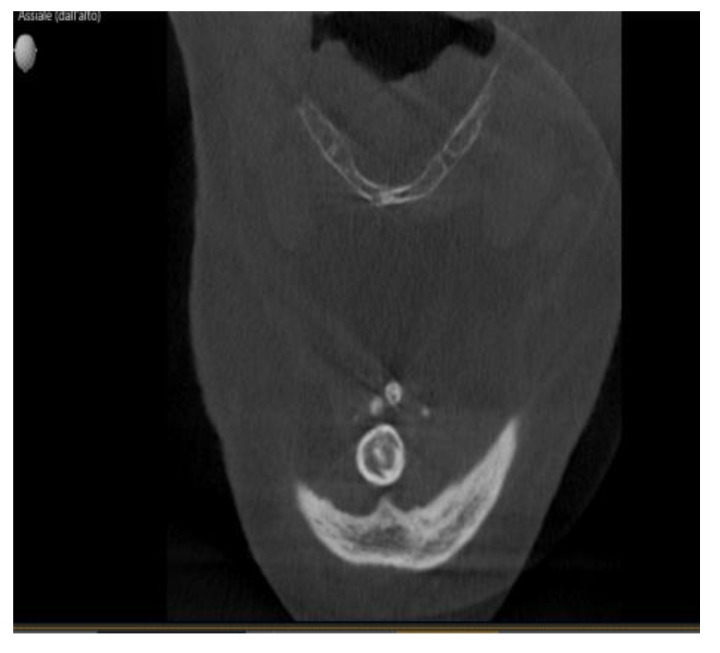
Same masses as in Figure 2 via a horizontal slice.

**Figure 4 diagnostics-13-00840-f004:**
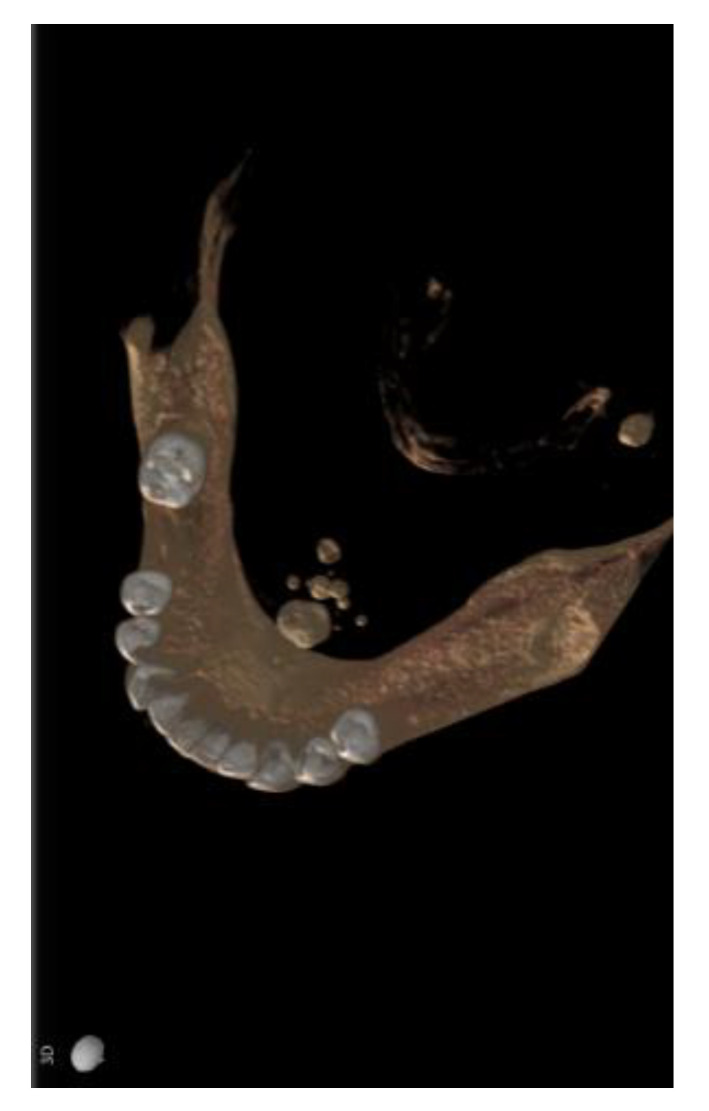
A 3D reconstruction base using CBCT information; radiographic findings also in submandibular area.

**Figure 5 diagnostics-13-00840-f005:**
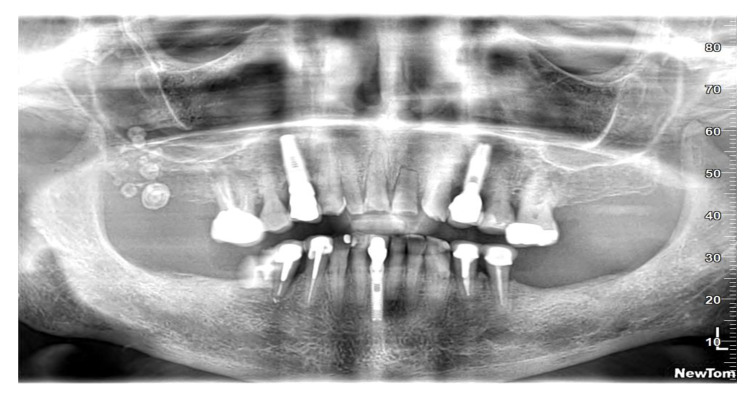
Orthopantomography of second patient MR: approximately a dozen oval/round masses could be observed in the right-half superior jaw.

**Figure 6 diagnostics-13-00840-f006:**
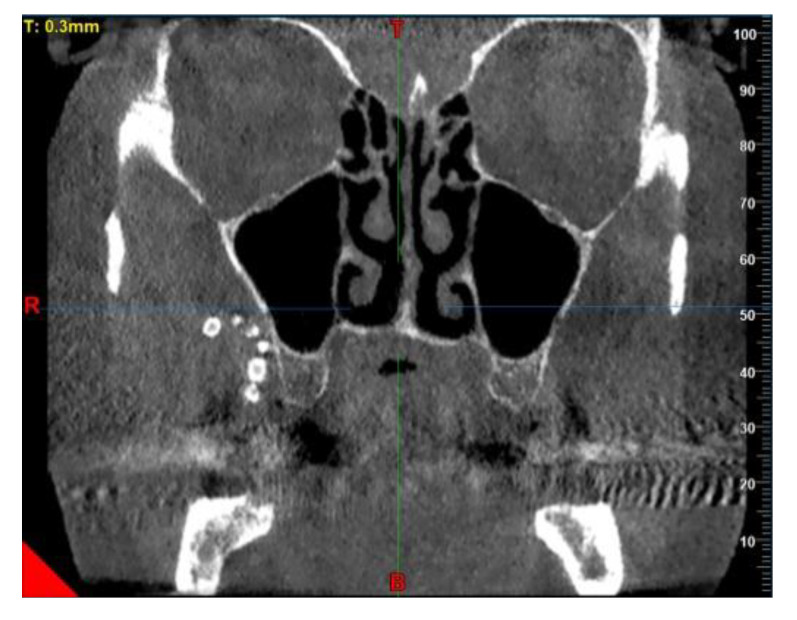
Sagittal slice of CBCT showing the enigmatic lesions in the tuber maxillae area.

**Figure 7 diagnostics-13-00840-f007:**
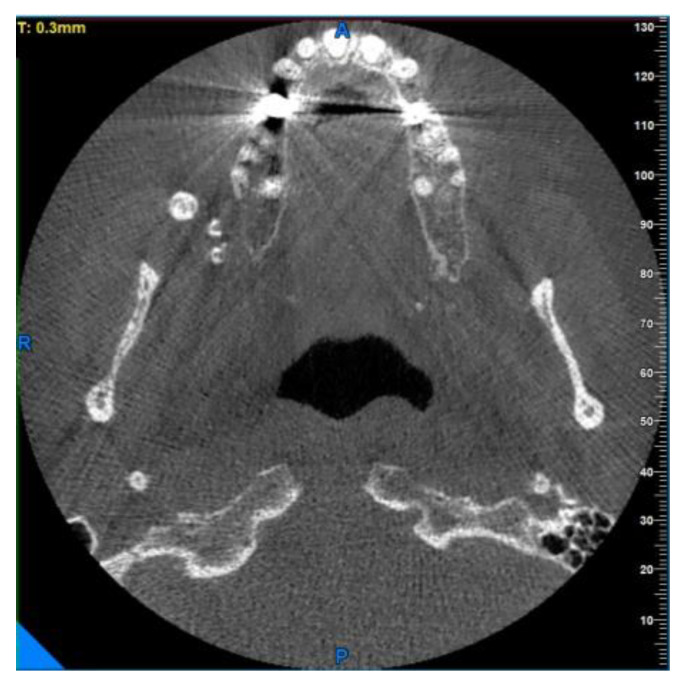
Horizontal slice with a well-defined 6.2 × 5.2 mm mass and others.

**Figure 8 diagnostics-13-00840-f008:**
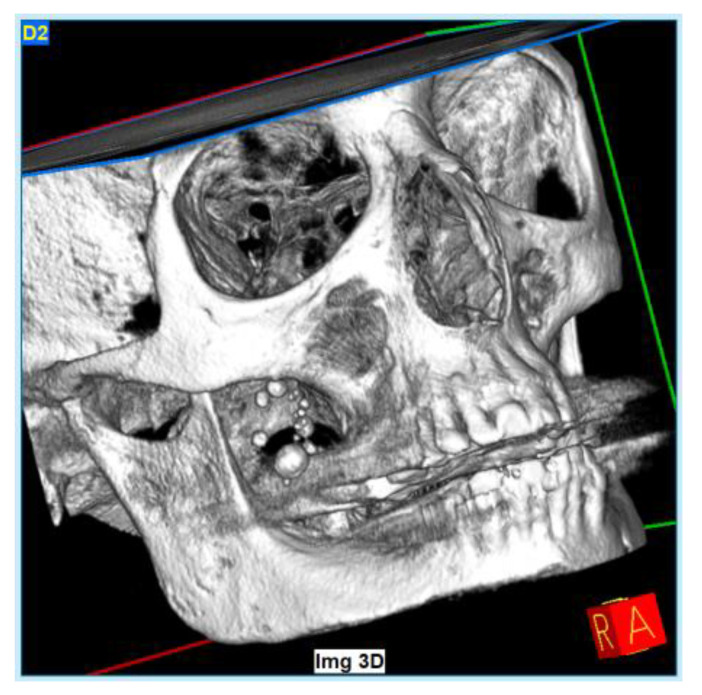
A 3D reconstruction using CBCT data.

**Table 1 diagnostics-13-00840-t001:** Main characteristics of the articles found.

Title	Author	Year of Publication	Contrast Medium	Number of Patients
Long-term retention of contrast medium in sialography: a case report [14].	A. El-Hadary, A. Ruprecht	1986	Lipiodol	1
Parotid and pterygomaxillary lipogranuloma caused by oil-based contrast medium used for sialography: report of a case [15].	Y. Shigetaka, S. Masatsugu, F. Yoshikuni, T. Yoshihiro	1996	Lipiodol	1
Lipiodol UF retention in dental sialography [16].	D. Ozdemir, N.T. Polat, S. Polat	2004	Lipiodol UF	1
Lipiodol ultra-fluid—foreign body in the cheek [17].	D. Macan, J. Hat, I. Luksic	2007	Lipiodol	1
Retention of lipiodol after parotid gland sialography [18].	J. Schortinghuis, J. Pijpe, F.K.L. Spijkervet, A. Vissink	2009	Lipiodol UF	28
Ethiodol extravasation during sialography [19].	P.N. Schilt, M.H. Fritsch	2010	Ethiodol	1

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
