# Peer review of "Enigmatic Formations Found in Routine Orthopantomography (OPG) Examinations: A Case Report"

_diagnostics, 2023, doi:10.3390/diagnostics13050840_

Round 1
Reviewer 1 Report
Dear authors,
I evaluated the article titled Enigmatic formations found in routine panorex examinations: a 1 case series of two patients and narrative review. There were many concerns about this article (below).
Title - it is wrong. There are no case series.
ABSTRACT - Incomplete. I suggest to rewrite it.
- The references were not correctly used.
- there is a table in the INTRO session which was developed to support data. I suggest to transcript it in the text.
- Only 2 refs in all INTRO are updated. Around 17 have more than 10 years. Update the literature accordingly.
- Case presentation: this part is incomplete.
- Discussion is partially adequate.
- Conclusion: it is inconclusive. Improve it.
Author Response
Dear reviewer, thank you for yourinterest and you precious suggestions you have given us in order to improve our paper.
- Title: we changed it and delete the wording "case series";
- Abstract: we rewrite it according with other reviewers indication;
- We transcript in the paper the intro's table, summarizing all the 6 articles mentioned;
- You are right: 17 references have more than 10 years, but unfortunatly as we emphasized several times in the text these types of diagnostic test were used much more in the past, now almost in disuse. This is the reason why we didn't find a lot of really important and significant recent papers to mention in the article;
- Conclusion: we developed it.
Thank you so much once again for your help.
Reviewer 2 Report
The manuscript has important discussion and articles that are related. As the authors explained, there are no recent manuscripts about this study so this can decrease the interest to the readers. Overall merit the article can be aproved
Author Response
Dear Reviewer,
thank you very much for appreciation shown to our manuscript.
We carefully checked everything few more times to find out misprints. In the attached you can find the corrected paper in its final form.
Thank you very much once again.

Reviewer 3 Report
This is a case series on two patients reporting occasional radiographic findings on OPG related to retention of contrast medium. The topic could be of interest, I would just arise the followings:
- For consistency I would use the term "Orthopantomography (OPG)" instead of "panorex" in all the text of the study including the title
- In the keywords I would use "CBCT" instead of "CTCB"
- Line 85 please revise writing "Tamac," instead of "Tamac ,"
- Line 97 please write the number 4 in apex
- Line 114 please correct "20106" with "2016"
- Please spell out the term CBCT (Cone Beam Computed Tomography) only the first time you mention it
Author Response
Dear Reviewer,
thank you very much for all the precious information you have provided us in order to improve our paper. We made the following changes (as indicated):
- We use the term "Orthopantomography (OPG)" instead of "panorex" in all the text of the study, first of all in the title;
- We changed from CTCB to CBCT (totally correct);
- We changed in line 85 to "Tamac," from "Tamac ,";
-
We wrote the number 4 in apex in line 97;
-
We corrected line 114 from "20106" to "2016" (misprint);
-
We spell out the term CBCT (Cone Beam Computed Tomography) the first time we mention it.
Thank you very much again for your help.
Round 2
Reviewer 1 Report
Dear authors,
I evaluated the article titled “Enigmatic formations found in routine Orthopantomography (OPG) examinations: two clinical cases and narrative review“. There were many concerns about this article (below).
Title - the authors need to define what kind of article it is. Review or case report.
ABSTRACT - Incomplete (how was done the review?). I suggest to rewrite it.
- Update the literature. The use of the refs are not correct.
- Case presentation: this part is incomplete.
- Discussion is partially correct.
Author Response
Dear reviewer,
thank you so much once again for your help.
We made the suggested corrections:
- Title: all authors agree that the article is a case report (the use of a narrative review in it was requested by another reviewer);
- Abstract: we partially re-write and completing it by inserting the modalities with which the narrative review was carried out;
- Update the literature: we have carefully corrected all references. As already mentioned, since the scialographic examination is now in disuse, there are not many significant papers that can be used and unfortunately they are dated. However, the most recent ones is from 2010;
- Case Presentation: could you kindly indicate in which section the case presentation is incomplete and what is specifically missing?
Thank you.
Best regards
Round 3
Reviewer 1 Report
Dear authors,
I evaluated the article titled “Enigmatic formations found in routine Orthopantomography (OPG) examinations: case report“. The article (case report) improved a lot; it has around 9 pages and it is more focused.
Thank you for the review.